# Learning a Probabilistic Latent Space of Object Shapes via 3D Generative-Adversarial Modeling

**Jiajun Wu***
MIT CSAIL

**Chengkai Zhang***
MIT CSAIL

**Tianfan Xue**
MIT CSAIL

**William T. Freeman**
MIT CSAIL, Google Research

**Joshua B. Tenenbaum**
MIT CSAIL

## Abstract

We study the problem of 3D object generation. We propose a novel framework, namely 3D Generative Adversarial Network (3D-GAN), which generates 3D objects from a probabilistic space by leveraging recent advances in volumetric convolutional networks and generative adversarial nets. The benefits of our model are three-fold: first, the use of an adversarial criterion, instead of traditional heuristic criteria, enables the generator to capture object structure implicitly and to synthesize high-quality 3D objects; second, the generator establishes a mapping from a low-dimensional probabilistic space to the space of 3D objects, so that we can sample objects without a reference image or CAD models, and explore the 3D object manifold; third, the adversarial discriminator provides a powerful 3D shape descriptor which, learned without supervision, has wide applications in 3D object recognition. Experiments demonstrate that our method generates high-quality 3D objects, and our unsupervisedly learned features achieve impressive performance on 3D object recognition, comparable with those of supervised learning methods.

## 1   Introduction

What makes a 3D generative model of object shapes appealing? We believe a good generative model should be able to synthesize 3D objects that are both highly varied and realistic. Specifically, for 3D objects to have variations, a generative model should be able to go beyond memorizing and recombining parts or pieces from a pre-defined repository to produce novel shapes; and for objects to be realistic, there need to be fine details in the generated examples.

In the past decades, researchers have made impressive progress on 3D object modeling and synthesis [Van Kaick et al., 2011, Tangelder and Veltkamp, 2008, Carlson, 1982], mostly based on meshes or skeletons. Many of these traditional methods synthesize new objects by borrowing parts from objects in existing CAD model libraries. Therefore, the synthesized objects look realistic, but not conceptually novel.

Recently, with the advances in deep representation learning and the introduction of large 3D CAD datasets like ShapeNet [Chang et al., 2015, Wu et al., 2015], there have been some inspiring attempts in learning deep object representations based on voxelized objects [Girdhar et al., 2016, Su et al., 2015a, Qi et al., 2016]. Different from part-based methods, many of these generative approaches do not explicitly model the concept of parts or retrieve them from an object repository; instead, they synthesize new objects based on learned object representations. This is a challenging problem because, compared to the space of 2D images, it is more difficult to model the space of 3D shapes due to its higher dimensionality. Their current results are encouraging, but often there still exist artifacts (*e.g.*, fragments or holes) in the generated objects.

In this paper, we demonstrate that modeling volumetric objects in a general-adversarial manner could be a promising solution to generate objects that are both novel and realistic. Our approach combines

---

∗ indicates equal contributions. Emails: {jiajunwu, ckzhang, tfxue, billf, jbt}@mit.edu

the merits of both general-adversarial modeling [Goodfellow et al., 2014, Radford et al., 2016] and volumetric convolutional networks [Maturana and Scherer, 2015, Wu et al., 2015]. Different from traditional heuristic criteria, generative-adversarial modeling introduces an adversarial discriminator to classify whether an object is synthesized or real. This could be a particularly favorable framework for 3D object modeling: as 3D objects are highly structured, a generative-adversarial criterion, but not a voxel-wise independent heuristic one, has the potential to capture the structural difference of two 3D objects. The use of a generative-adversarial loss may also avoid possible criterion-dependent overfitting (*e.g.*, generating mean-shape-like blurred objects when minimizing a mean squared error).

Modeling 3D objects in a generative-adversarial way offers additional distinctive advantages. First, it becomes possible to sample novel 3D objects from a probabilistic latent space such as a Gaussian or uniform distribution. Second, the discriminator in the generative-adversarial approach carries informative features for 3D object recognition, as demonstrated in experiments (Section 4). From a different perspective, instead of learning a single feature representation for both generating and recognizing objects [Girdhar et al., 2016, Sharma et al., 2016], our framework learns disentangled generative and discriminative representations for 3D objects without supervision, and applies them on generation and recognition tasks, respectively.

We show that our generative representation can be used to synthesize high-quality realistic objects, and our discriminative representation can be used for 3D object recognition, achieving comparable performance with recent supervised methods [Maturana and Scherer, 2015, Shi et al., 2015], and outperforming other unsupervised methods by a large margin. The learned generative and discriminative representations also have wide applications. For example, we show that our network can be combined with a variational autoencoder [Kingma and Welling, 2014, Larsen et al., 2016] to directly reconstruct a 3D object from a 2D input image. Further, we explore the space of object representations and demonstrate that both our generative and discriminative representations carry rich semantic information about 3D objects.

## 2 Related Work

**Modeling and synthesizing 3D shapes** 3D object understanding and generation is an important problem in the graphics and vision community, and the relevant literature is very rich [Carlson, 1982, Tangelder and Veltkamp, 2008, Van Kaick et al., 2011, Blanz and Vetter, 1999, Kalogerakis et al., 2012, Chaudhuri et al., 2011, Xue et al., 2012, Kar et al., 2015, Bansal et al., 2016, Wu et al., 2016]. Since decades ago, AI and vision researchers have made inspiring attempts to design or learn 3D object representations, mostly based on meshes and skeletons. Many of these shape synthesis algorithms are nonparametric and they synthesize new objects by retrieving and combining shapes and parts from a database. Recently, Huang et al. [2015] explored generating 3D shapes with pre-trained templates and producing both object structure and surface geometry. Our framework synthesizes objects without explicitly borrow parts from a repository, and requires no supervision during training.

**Deep learning for 3D data** The vision community have witnessed rapid development of deep networks for various tasks. In the field of 3D object recognition, Li et al. [2015], Su et al. [2015b], Girdhar et al. [2016] proposed to learn a joint embedding of 3D shapes and synthesized images, Su et al. [2015a], Qi et al. [2016] focused on learning discriminative representations for 3D object recognition, Wu et al. [2016], Xiang et al. [2015], Choy et al. [2016] discussed 3D object reconstruction from in-the-wild images, possibly with a recurrent network, and Girdhar et al. [2016], Sharma et al. [2016] explored autoencoder-based networks for learning voxel-based object representations. Wu et al. [2015], Rezende et al. [2016], Yan et al. [2016] attempted to generate 3D objects with deep networks, some using 2D images during training with a 3D to 2D projection layer. Many of these networks can be used for 3D shape classification [Su et al., 2015a, Sharma et al., 2016, Maturana and Scherer, 2015], 3D shape retrieval [Shi et al., 2015, Su et al., 2015a], and single image 3D reconstruction [Kar et al., 2015, Bansal et al., 2016, Girdhar et al., 2016], mostly with full supervision. In comparison, our framework requires no supervision for training, is able to generate objects from a probabilistic space, and comes with a rich discriminative 3D shape representation.

**Learning with an adversarial net** Generative Adversarial Nets (GAN) [Goodfellow et al., 2014] proposed to incorporate an adversarial discriminator into the procedure of generative modeling. More recently, LAPGAN [Denton et al., 2015] and DC-GAN [Radford et al., 2016] adopted GAN with convolutional networks for image synthesis, and achieved impressive performance. Researchers have also explored the use of GAN for other vision problems. To name a few, Wang and Gupta [2016] discussed how to model image style and structure with sequential GANs, Li and Wand [2016] and Zhu et al. [2016] used GAN for texture synthesis and image editing, respectively, and Im et al. [2016]

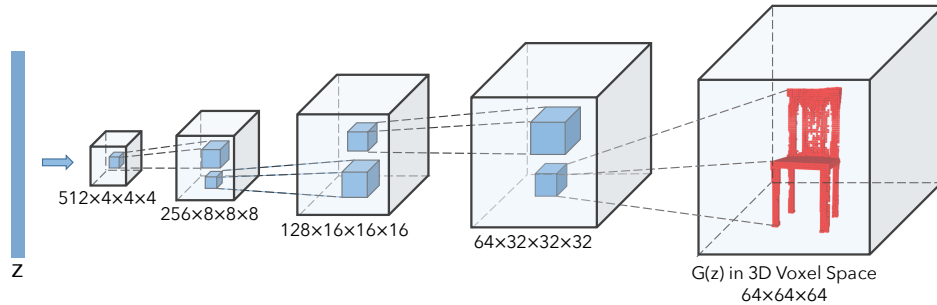

Figure 1: The generator in 3D-GAN. The discriminator mostly mirrors the generator.

developed a recurrent adversarial network for image generation. While previous approaches focus on modeling 2D images, we discuss the use of an adversarial component in modeling 3D objects.

## 3 Models

In this section we introduce our model for 3D object generation. We first discuss how we build our framework, 3D Generative Adversarial Network (3D-GAN), by leveraging previous advances on volumetric convolutional networks and generative adversarial nets. We then show how to train a variational autoencoder [Kingma and Welling, 2014] simultaneously so that our framework can capture a mapping from a 2D image to a 3D object.

### 3.1 3D Generative Adversarial Network (3D-GAN)

As proposed in Goodfellow et al. [2014], the Generative Adversarial Network (GAN) consists of a generator and a discriminator, where the discriminator tries to classify real objects and objects synthesized by the generator, and the generator attempts to confuse the discriminator. In our 3D Generative Adversarial Network (3D-GAN), the generator $G$ maps a 200-dimensional latent vector $z$, randomly sampled from a probabilistic latent space, to a $64 \times 64 \times 64$ cube, representing an object $G(z)$ in 3D voxel space. The discriminator $D$ outputs a confidence value $D(x)$ of whether a 3D object input $x$ is real or synthetic.

Following Goodfellow et al. [2014], we use binary cross entropy as the classification loss, and present our overall adversarial loss function as

$$L_{\text{3D-GAN}} = \log D(x) + \log(1 - D(G(z))), \tag{1}$$

where $x$ is a real object in a $64 \times 64 \times 64$ space, and $z$ is a randomly sampled noise vector from a distribution $p(z)$. In this work, each dimension of $z$ is an i.i.d. uniform distribution over $[0, 1]$.

**Network structure** Inspired by Radford et al. [2016], we design an all-convolutional neural network to generate 3D objects. As shown in Figure 1, the generator consists of five volumetric fully convolutional layers of kernel sizes $4 \times 4 \times 4$ and strides 2, with batch normalization and ReLU layers added in between and a Sigmoid layer at the end. The discriminator basically mirrors the generator, except that it uses Leaky ReLU [Maas et al., 2013] instead of ReLU layers. There are no pooling or linear layers in our network. More details can be found in the supplementary material.

**Training details** A straightforward training procedure is to update both the generator and the discriminator in every batch. However, the discriminator usually learns much faster than the generator, possibly because generating objects in a 3D voxel space is more difficult than differentiating between real and synthetic objects [Goodfellow et al., 2014, Radford et al., 2016]. It then becomes hard for the generator to extract signals for improvement from a discriminator that is way ahead, as all examples it generated would be correctly identified as synthetic with high confidence. Therefore, to keep the training of both networks in pace, we employ an adaptive training strategy: for each batch, the discriminator only gets updated if its accuracy in the last batch is not higher than 80%. We observe this helps to stabilize the training and to produce better results. We set the learning rate of $G$ to 0.0025, $D$ to $10^{-5}$, and use a batch size of 100. We use ADAM [Kingma and Ba, 2015] for optimization, with $\beta = 0.5$.

### 3.2 3D-VAE-GAN

We have discussed how to generate 3D objects by sampling a latent vector $z$ and mapping it to the object space. In practice, it would also be helpful to infer these latent vectors from observations. For example, if there exists a mapping from a 2D image to the latent representation, we can then recover the 3D object corresponding to that 2D image.

Following this idea, we introduce 3D-VAE-GAN as an extension to 3D-GAN. We add an additional image encoder $E$, which takes a 2D image $x$ as input and outputs the latent representation vector $z$. This is inspired by VAE-GAN proposed by [Larsen et al., 2016], which combines VAE and GAN by sharing the decoder of VAE with the generator of GAN.

The 3D-VAE-GAN therefore consists of three components: an image encoder $E$, a decoder (the generator $G$ in 3D-GAN), and a discriminator $D$. The image encoder consists of five spatial convolution layers with kernel size $\{11, 5, 5, 5, 8\}$ and strides $\{4, 2, 2, 2, 1\}$, respectively. There are batch normalization and ReLU layers in between, and a sampler at the end to sample a 200 dimensional vector used by the 3D-GAN. The structures of the generator and the discriminator are the same as those in Section 3.1.

Similar to VAE-GAN [Larsen et al., 2016], our loss function consists of three parts: an object reconstruction loss $L_{\text{recon}}$, a cross entropy loss $L_{\text{3D-GAN}}$ for 3D-GAN, and a KL divergence loss $L_{\text{KL}}$ to restrict the distribution of the output of the encoder. Formally, these loss functions write as

$$L = L_{\text{3D-GAN}} + \alpha_1 L_{\text{KL}} + \alpha_2 L_{\text{recon}}, \tag{2}$$

where $\alpha_1$ and $\alpha_2$ are weights of the KL divergence loss and the reconstruction loss. We have

$$L_{\text{3D-GAN}} = \log D(x) + \log(1 - D(G(z))), \tag{3}$$

$$L_{\text{KL}} = D_{\text{KL}}(q(z|y) \ || \ p(z)), \tag{4}$$

$$L_{\text{recon}} = ||G(E(y)) - x||_2, \tag{5}$$

where $x$ is a 3D shape from the training set, $y$ is its corresponding 2D image, and $q(z|y)$ is the variational distribution of the latent representation $z$. The KL-divergence pushes this variational distribution towards to the prior distribution $p(z)$, so that the generator can sample the latent representation $z$ from the same distribution $p(z)$. In this work, we choose $p(z)$ a multivariate Gaussian distribution with zero-mean and unit variance. For more details, please refer to Larsen et al. [2016].

Training 3D-VAE-GAN requires both 2D images and their corresponding 3D models. We render 3D shapes in front of background images ($16,913$ indoor images from the SUN database [Xiao et al., 2010]) in 72 views (from 24 angles and 3 elevations). We set $\alpha_1 = 5$, $\alpha_2 = 10^{-4}$, and use a similar training strategy as in Section 3.1. See our supplementary material for more details.

## 4   Evaluation

In this section, we evaluate our framework from various aspects. We first show qualitative results of generated 3D objects. We then evaluate the unsupervisedly learned representation from the discriminator by using them as features for 3D object classification. We show both qualitative and quantitative results on the popular benchmark ModelNet [Wu et al., 2015]. Further, we evaluate our 3D-VAE-GAN on 3D object reconstruction from a single image, and show both qualitative and quantitative results on the IKEA dataset [Lim et al., 2013].

### 4.1   3D Object Generation

Figure 2 shows 3D objects generated by our 3D-GAN. For this experiment, we train one 3D-GAN for each object category. For generation, we sample 200-dimensional vectors following an i.i.d. uniform distribution over $[0, 1]$, and render the largest connected component of each generated object. We compare 3D-GAN with Wu et al. [2015], the state-of-the-art in 3D object synthesis from a probabilistic space, and with a volumetric autoencoder, whose variants have been employed by multiple recent methods [Girdhar et al., 2016, Sharma et al., 2016]. Because an autoencoder does not restrict the distribution of its latent representation, we compute the empirical distribution $p_0(z)$ of the latent vector $z$ of all training examples, fit a Gaussian distribution $g_0$ to $p_0$, and sample from $g_0$. Our algorithm produces 3D objects with much higher quality and more fine-grained details.

Compared with previous works, our 3D-GAN can synthesize high-resolution 3D objects with detailed geometries. Figure 3 shows both high-res voxels and down-sampled low-res voxels for comparison. Note that it is relatively easy to synthesize a low-res object, but is much harder to obtain a high-res one due to the rapid growth of 3D space. However, object details are only revealed in high resolution.

A natural concern to our generative model is whether it is simply memorizing objects from training data. To demonstrate that the network can generalize beyond the training set, we compare synthesized objects with their nearest neighbor in the training set. Since the retrieval objects based on $\ell^2$ distance in the voxel space are visually very different from the queries, we use the output of the last convolutional

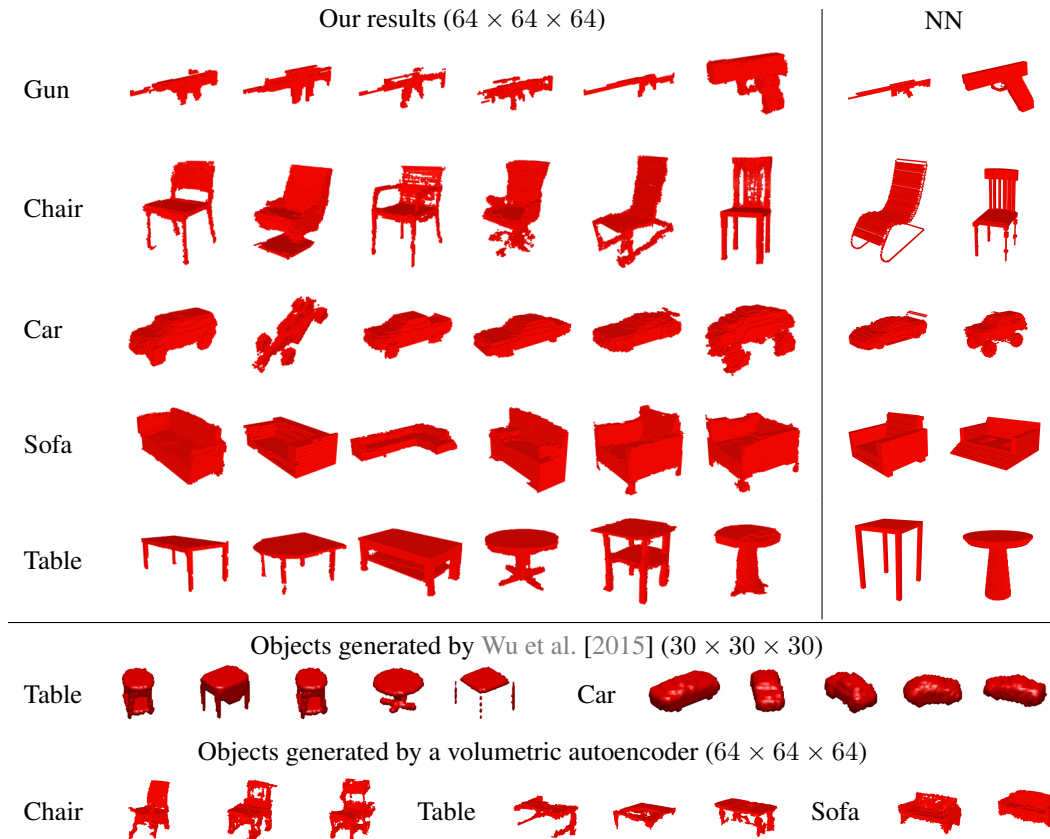

Figure 2: Objects generated by 3D-GAN from vectors, without a reference image/object. We show, for the last two objects in each row, the nearest neighbor retrieved from the training set. We see that the generated objects are similar, but not identical, to examples in the training set. For comparison, we show objects generated by the previous state-of-the-art [Wu et al., 2015] (results supplied by the authors). We also show objects generated by autoencoders trained on a single object category, with latent vectors sampled from empirical distribution. See text for details.

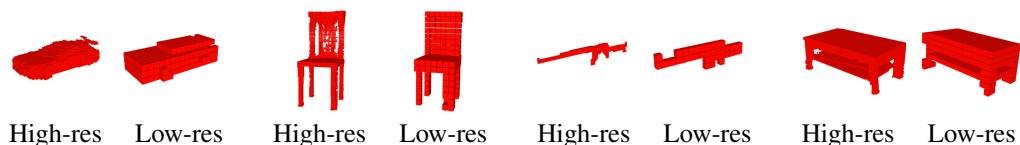

Figure 3: We present each object at high resolution ($64 \times 64 \times 64$) on the left and at low resolution (down-sampled to $16 \times 16 \times 16$) on the right. While humans can perceive object structure at a relatively low resolution, fine details and variations only appear in high-res objects.

layer in our discriminator (with a 2x pooling) as features for retrieval instead. Figure 2 shows that generated objects are similar, but not identical, to the nearest examples in the training set.

## 4.2 3D Object Classification

We then evaluate the representations learned by our discriminator. A typical way of evaluating representations learned without supervision is to use them as features for classification. To obtain features for an input 3D object, we concatenate the responses of the second, third, and fourth convolution layers in the discriminator, and apply max pooling of kernel sizes $\{8, 4, 2\}$, respectively. We use a linear SVM for classification.

**Data** We train a single 3D-GAN on the seven major object categories (chairs, sofas, tables, boats, airplanes, rifles, and cars) of ShapeNet [Chang et al., 2015]. We use ModelNet [Wu et al., 2015] for testing, following Sharma et al. [2016], Maturana and Scherer [2015], Qi et al. [2016].* Specifically, we evaluate our model on both ModelNet10 and ModelNet40, two subsets of ModelNet that are often

| Supervision | Pretraining | Method | Classification (Accuracy) | |
|---|---|---|---|---|
| | | | ModelNet40 | ModelNet10 |
| Category labels | ImageNet | MVCNN [Su et al., 2015a] | 90.1% | - |
| | | MVCNN-MultiRes [Qi et al., 2016] | **91.4**% | - |
| | None | 3D ShapeNets [Wu et al., 2015] | 77.3% | 83.5% |
| | | DeepPano [Shi et al., 2015] | 77.6% | 85.5% |
| | | VoxNet [Maturana and Scherer, 2015] | 83.0% | 92.0% |
| | | ORION [Sedaghat et al., 2016] | - | **93.8**% |
| Unsupervised | - | SPH [Kazhdan et al., 2003] | 68.2% | 79.8% |
| | | LFD [Chen et al., 2003] | 75.5% | 79.9% |
| | | T-L Network [Girdhar et al., 2016] | 74.4% | - |
| | | VConv-DAE [Sharma et al., 2016] | 75.5% | 80.5% |
| | | 3D-GAN (ours) | **83.3**% | **91.0**% |

Table 1: Classification results on the ModelNet dataset. Our 3D-GAN outperforms other unsupervised learning methods by a large margin, and is comparable to some recent supervised learning frameworks.

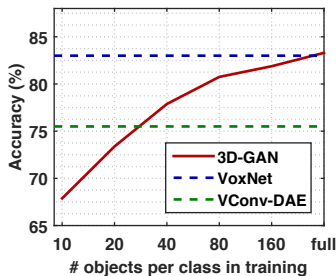

Figure 4: ModelNet40 classification with limited training data

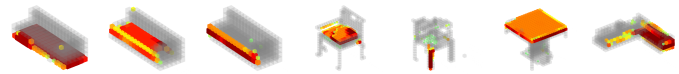

Figure 5: The effects of individual dimensions of the object vector

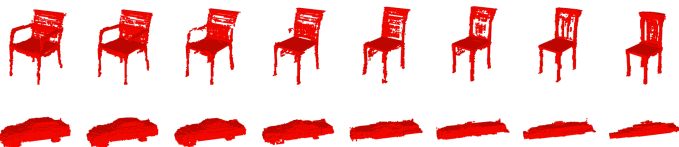

Figure 6: Intra/inter-class interpolation between object vectors

used as benchmarks for 3D object classification. Note that the training and test categories are not identical, which also shows the out-of-category generalization power of our 3D-GAN.

**Results** We compare with the state-of-the-art methods [Wu et al., 2015, Girdhar et al., 2016, Sharma et al., 2016, Sedaghat et al., 2016] and show per-class accuracy in Table 1. Our representation outperforms other features learned without supervision by a large margin (83.3% vs. 75.5% on ModelNet40, and 91.0% vs 80.5% on ModelNet10) [Girdhar et al., 2016, Sharma et al., 2016]. Further, our classification accuracy is also higher than some recent supervised methods [Shi et al., 2015], and is close to the state-of-the-art voxel-based supervised learning approaches [Maturana and Scherer, 2015, Sedaghat et al., 2016]. Multi-view CNNs [Su et al., 2015a, Qi et al., 2016] outperform us, though their methods are designed for classification, and require rendered multi-view images and an ImageNet-pretrained model.

3D-GAN also works well with limited training data. As shown in Figure 4, with roughly 25 training samples per class, 3D-GAN achieves comparable performance on ModelNet40 with other unsupervised learning methods trained with at least 80 samples per class.

### 4.3 Single Image 3D Reconstruction

As an application, our show that the 3D-VAE-GAN can perform well on single image 3D reconstruction. Following previous work [Girdhar et al., 2016], we test it on the IKEA dataset [Lim et al., 2013], and show both qualitative and quantitative results.

**Data** The IKEA dataset consists of images with IKEA objects. We crop the images so that the objects are centered in the images. Our test set consists of 1,039 objects cropped from 759 images (supplied by the author). The IKEA dataset is challenging because all images are captured in the wild, often with heavy occlusions. We test on all six categories of objects: bed, bookcase, chair, desk, sofa, and table.

**Results** We show our results in Figure 7 and Table 2, with performance of a single 3D-VAE-GAN jointly trained on all six categories, as well as the results of six 3D-VAE-GANs separately trained on

---

et al. [2015a], Sharma et al. [2016] used 80 training points and 20 test points in each category for experiments, possibly with viewpoint augmentation.

| Method | Bed | Bookcase | Chair | Desk | Sofa | Table | Mean |
|---|---|---|---|---|---|---|---|
| AlexNet-fc8 [Girdhar et al., 2016] | 29.5 | 17.3 | 20.4 | 19.7 | 38.8 | 16.0 | 23.6 |
| AlexNet-conv4 [Girdhar et al., 2016] | 38.2 | 26.6 | 31.4 | 26.6 | 69.3 | 19.1 | 35.2 |
| T-L Network [Girdhar et al., 2016] | 56.3 | 30.2 | 32.9 | 25.8 | 71.7 | 23.3 | 40.0 |
| 3D-VAE-GAN (jointly trained) | 49.1 | 31.9 | 42.6 | 34.8 | **79.8** | 33.1 | 45.2 |
| 3D-VAE-GAN (separately trained) | **63.2** | **46.3** | **47.2** | **40.7** | 78.8 | **42.3** | **53.1** |

Table 2: Average precision for voxel prediction on the IKEA dataset.[†]

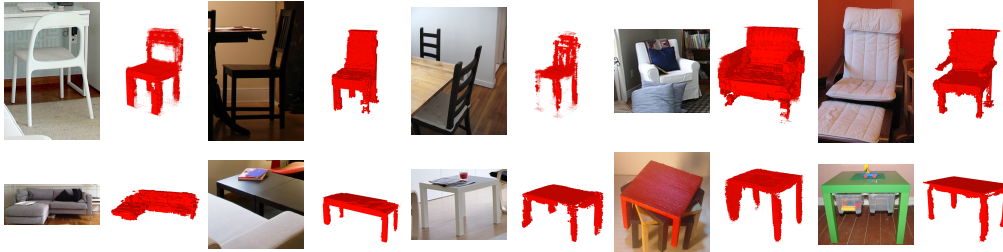

Figure 7: Qualitative results of single image 3D reconstruction on the IKEA dataset

each class. Following Girdhar et al. [2016], we evaluate results at resolution $20 \times 20 \times 20$, use the average precision as our evaluation metric, and attempt to align each prediction with the ground-truth over permutations, flips, and translational alignments (up to 10%), as IKEA ground truth objects are not in a canonical viewpoint. In all categories, our model consistently outperforms previous state-of-the-art in voxel-level prediction and other baseline methods.[†]

## 5 Analyzing Learned Representations

In this section, we look deep into the representations learned by both the generator and the discriminator of 3D-GAN. We start with the 200-dimensional object vector, from which the generator produces various objects. We then visualize neurons in the discriminator, and demonstrate that these units capture informative semantic knowledge of the objects, which justifies its good performance on object classification presented in Section 4.

### 5.1 The Generative Representation

We explore three methods for understanding the latent space of vectors for object generation. We first visualize what an individual dimension of the vector represents; we then explore the possibility of interpolating between two object vectors and observe how the generated objects change; last, we present how we can apply shape arithmetic in the latent space.

**Visualizing the object vector** To visualize the semantic meaning of each dimension, we gradually increase its value, and observe how it affects the generated 3D object. In Figure 5, each column corresponds to one dimension of the object vector, where the red region marks the voxels affected by changing values of that dimension. We observe that some dimensions in the object vector carries semantic knowledge of the object, *e.g.*, the thickness or width of surfaces.

**Interpolation** We show results of interpolating between two object vectors in Figure 6. Earlier works demonstrated interpolation between two 2D images of the same category [Dosovitskiy et al., 2015, Radford et al., 2016]. Here we show interpolations both within and across object categories. We observe that for both cases walking over the latent space gives smooth transitions between objects.

**Arithmetic** Another way of exploring the learned representations is to show arithmetic in the latent space. Previously, Dosovitskiy et al. [2015], Radford et al. [2016] presented that their generative nets are able to encode semantic knowledge of chair or face images in its latent space; Girdhar et al. [2016] also showed that the learned representation for 3D objects behave similarly. We show our shape arithmetic in Figure 8. Different from Girdhar et al. [2016], all of our objects are randomly sampled, requiring no existing 3D CAD models as input.

### 5.2 The Discriminative Representation

We now visualize the neurons in the discriminator. Specifically, we would like to show what input objects, and which part of them produce the highest intensity values for each neuron. To do that,

---

[†]For methods from Girdhar et al. [2016], the mean values in the last column are higher than the originals in their paper, because we compute per-class accuracy instead of per-instance accuracy.

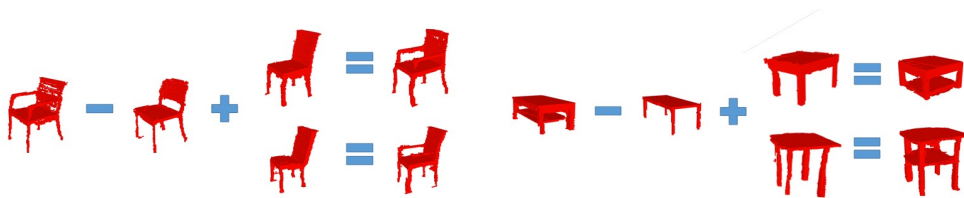

Figure 8: Shape arithmetic for chairs and tables. The left images show the obtained "arm" vector can be added to other chairs, and the right ones show the "layer" vector can be added to other tables.

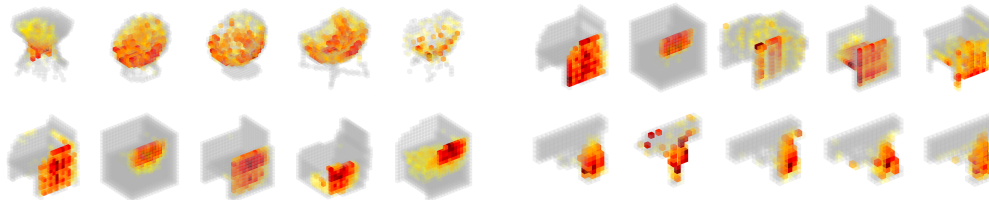

Figure 9: Objects and parts that activate specific neurons in the discriminator. For each neuron, we show five objects that activate it most strongly, with colors representing gradients of activations with respect to input voxels.

for each neuron in the second to last convolutional layer of the discriminator, we iterate through all training objects and exhibit the ones activating the unit most strongly. We further use guided back-propagation [Springenberg et al., 2015] to visualize the parts that produce the activation.

Figure 9 shows the results. There are two main observations: first, for a single neuron, the objects producing strongest activations have very similar shapes, showing the neuron is selective in terms of the overall object shape; second, the parts that activate the neuron, shown in red, are consistent across these objects, indicating the neuron is also learning semantic knowledge about object parts.

# 6 Conclusion

In this paper, we proposed 3D-GAN for 3D object generation, as well as 3D-VAE-GAN for learning an image to 3D model mapping. We demonstrated that our models are able to generate novel objects and to reconstruct 3D objects from images. We showed that the discriminator in GAN, learned without supervision, can be used as an informative feature representation for 3D objects, achieving impressive performance on shape classification. We also explored the latent space of object vectors, and presented results on object interpolation, shape arithmetic, and neuron visualization.

**Acknowledgement**    This work is supported by NSF grants #1212849 and #1447476, ONR MURI N00014-16-1-2007, the Center for Brain, Minds and Machines (NSF STC award CCF-1231216), Toyota Research Institute, Adobe, Shell, IARPA MICrONS, and a hardware donation from Nvidia.

## Footnotes

*For ModelNet, there are two train/test splits typically used. Qi et al. [2016], Shi et al. [2015], Maturana and Scherer [2015] used the train/test split included in the dataset, which we also follow; Wu et al. [2015], Su

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
