[Supplementary Material · 0062_supp.pdf]

## A.1 Network Structure

Here we give the network structures of the generator, the discriminator, and the image encoder.

**Generator** The generator consists of five fully convolution layers with numbers of channels $\{512, 256, 128, 64, 1\}$, kernel sizes $\{4, 4, 4, 4, 4\}$, and strides $\{1, 2, 2, 2, 2\}$. We add ReLU and batch normalization layers between convolutional layers, and a Sigmoid layer at the end. The input is a 200-dimensional vector, and the output is a $64 \times 64 \times 64$ matrix with values in $[0, 1]$.

**Discriminator** As a mirrored version of the generator, the discriminator takes as input a $64 \times 64 \times 64$ matrix, and outputs a real number in $[0, 1]$. The discriminator consists of 5 volumetric convolution layers, with numbers of channels $\{64, 128, 256, 512, 1\}$, kernel sizes $\{4, 4, 4, 4, 4\}$, and strides $\{2, 2, 2, 2, 1\}$. There are leaky ReLU layers of parameter 0.2 and batch normalization layers in between, and a Sigmoid layer at the end.

**Image encoder** The image encoder in our 3D-VAE-GAN takes a $3 \times 256 \times 256$ image as input, and outputs a 200-dimensional vector. It consists of five spatial convolution layers with numbers of channels $\{64, 128, 256, 512, 400\}$, kernel sizes $\{11, 5, 5, 5, 8\}$, and strides $\{4, 2, 2, 2, 1\}$, respectively. There are ReLU and batch normalization layers in between. The output of the last convolution layer is a 400-dimensional vector representing a Gaussian distribution in the 200-dimensional space, where 200 dimensions are for the mean and the other 200 dimensions are for the diagonal variance. There is a sampling layer at the end to sample a 200-dimensional vector from the Gaussian distribution, which is later used by the 3D-GAN.

## A.2 3D Shape Classification

Here we present the details of our shape classification experiments. For each object, we take the responses of the second, third, and fourth convolution layers of the discriminator, and apply max pooling of kernel sizes $\{8, 4, 2\}$, respectively. We then concatenate the outputs into a vector of length 7,168, which is later used by a linear SVM for training and classification.

We use one-versus-all SVM classifiers. We use L2 penalty, balanced class weights, and intercept scaling during training. For ModelNet40, we train a linear SVM with penalty parameter $C = 0.07$, and for ModelNet10, we have $C = 0.01$. We also show the results with limited training data on both ModelNet40 and ModelNet10 in Figure A1.

Figure A1: Classification accuracy with limited training data, on ModelNet40 and ModelNet10

## A.3 3D-VAE-GAN Training

Let $\{x_i, y_i\}$ be the set of training pairs, where $y_i$ is a 2D image and $x_i$ is the corresponding 3D shape. In each iteration $t$ of training, we first generate a random sample $z_t$ from $N(\mathbf{0}, \mathbf{I})^*$. Then we update the discriminator $D$, the image encoder $E$, and the generator $G$ sequentially. Specifically,

---

$^*\mathbf{I}$ is an identity matrix.

- Step 1: Update the discriminator $D$ by minimizing the following loss function:

$$\log D(x_i) + \log(1 - D(G(z_t))). \tag{1}$$

- Step 2: Update the image encoder $E$ by minimizing the following loss function:

$$D_{\mathrm{KL}}\left(N(E_{\mathrm{mean}}(y_i), E_{\mathrm{var}}(y_i)) \parallel N(\mathbf{0}, \mathbf{I})\right) + ||G(E(y_i)) - x_i||_2, \tag{2}$$

where $E_{\mathrm{mean}}(y_i)$ and $E_{\mathrm{var}}(y_i)$ are the predicted mean and variance of the latent variable $z$, respectively.

- Step 3: Update the generator $G$ by minimizing the following loss function:

$$\log(1 - D(G(z_t))) + ||G(E(y_i)) - x_i||_2. \tag{3}$$