[Reviews · NeurIPS 2016]

Reviewer 1

Summary

The paper applies both generative adversarial networks (GAN) and variational auto encoders (VAE) to volumetric data. This * enables modeling and sampling the distribution of 3d volumetric shapes, * yields a 3d shape descriptor, and * yields a mapping from (2d) photos to the 3d shape descriptor and thus a 3d shape reconstruction without supervision (only training data are voxel grids). The paper evaluates 3d object classification on ModelNet and single image 3d reconstruction on the IKEA dataset. It shows qualitative results for samples from the 3d shape distribution, effect of different dimensions of the 3d shape descriptor, 3d shape interpolation, 3d shape arithmetic, and discriminative features learned by the network.

Qualitative Assessment

The novelty is somewhat limited: The combination of VAE and GAN to enable the presented tasks has been used in this constellation in Larsen et al. 2016. New is the extension to volumetric data. This is achieved with architectures similar to the ones used in Radford et al. 2016, Sharma et al. 2016. The extensions appears relatively straight forward, ie. volumetric convolutions. Quantitative results are convincing and show significant improvements. For 3d shape classification the presented method is competitive to fully supervised related work (ignoring the methods that are pretrained on imagenet) even though in this work the representation learning is unsupervised. Compared to other unsupervised methods, this method is stronger by quite a gap, halving the number of mistakes on ModelNet10. For single image 3d reconstruction this method also shows a significant improvement from 38% to 53% accuracy on the IKEA benchmark. The paper is well written, presented clearly, and contains a decent amount of technical details (as in details of how to train the model). References are adequate. (There are no proofs or theoretic analyses.) There are several qualitative results, giving some analysis of the model and it's quality. Sometimes it is a bit unclear what the insight is. Overall, although the novelty is limited, the paper demonstrates (1) how well the representation learnt by GANs and VAEs performs and (2) significantly advances the state of the art for unsupervised representations in 3d volumetric shape classification and single image 3d reconstruction. These insights will be interesting to the community.

Confidence in this Review

2-Confident (read it all; understood it all reasonably well)


Reviewer 2

Summary

The paper proposes a generative adversarial networks (GAN) framework for 3D object generation, which is called the Volumetric Adversarial Networks (VAN). In order to adapt the GAN framework for the 3D object generation task, 3D volumetric convolutional architecture is used for the the generator and discriminator. The paper also combines the VAN network with a variational autoencoder for the purpose of synthesizing 3D shape from 2D query image, which is termed as VAE-VAN. A set of experiment results were presented for evaluating the proposed method, which include 1) visual comparison of the generated 3D shape with a prior work, 2) shape classification performance using the unsupervised feature extracted from the discriminator network with several prior works 3) visualization of the generated 3D objects from color images using the VAE-VAN framework, and 4) quantitative comparison of the generated 3D objects from color images using the VAE-VAN framework with several works. The paper also shows the shape arithmetic operation enabled by the VAN and visualizes the neurons in the discriminator.

Qualitative Assessment

In terms of novelty, the paper is in many ways a straight forward extension of the GAN and DCGAN works to the 3D object generation task by replacing the 2D image generation and discrimination networks with 3D volumetric generation and discrimination networks. The major contribution of the paper is empirically showing that several nice properties of the GAN and DCGAN for the 2D image generation task also exhibit for the 3D object generation task. The properties include the capability of generating novel objects and the strength of the discriminator network as a feature extractor for the classification task. In terms of potential impact, the paper shows that the unsupervised features from the discriminator network are more discriminative than the other unsupervisedly learned features but are only as discriminative as the state-of-the-art algorithms that train their feature extractors supervisedly. As training a classification layer for classifying the unsupervisedly learned features extracted from the discriminator network, the labels of the samples have to be used. This does not reduce the number of samples required for achieving a target performance. It is hence unclear the advantage of the proposed unsupervised feature learning approach. It would be more convincing if the proposed method can achieve comparable or better performance with a fewer number of labeled examples using the proposed unsupervised features. The paper is easy to follow. I do not have difficulty in understanding the paper.

Confidence in this Review

3-Expert (read the paper in detail, know the area, quite certain of my opinion)


Reviewer 3

Summary

The authors propose a latent representation of voxelized volumes suitable both for generative and discriminative purposes. This representation is also linked to cropped 2D images. The system is based on Generative Adversarial Nets (GANs, Goodfellow et al. 2014), where the generative net is replaced by an VAE encoder, as in Larsen et al. 2016. The present manuscript focuses on 3D volume data, as opposed to 2D images from those two publications. The method shows qualitative and quantitative experiments related to 3D object generation, 3D object classification and 3D reconstruction from RGB images, with good results compared to the state of the art.

Qualitative Assessment

The present paper shows a nice application of the system presented in Larsen 2016 to 3D volumetric data. Previous work (well analyzed in the manuscript) on volumetric data focuses on combining existing parts, creating latent spaces by exploiting class labels or simple losses in voxel space. The authors use the research on VAE and GAN (Larsen 2016) on this type of data, adapting the convolutional layers to volumetric ones, to improve the performance of the state-of-the-art in this type of data. The experiments showcase three potential interesting uses: object generation, 3D object classification, and 3D objects from RGB images. The analysis of the latent representation in section 5 is also interesting. On the negative side, there's basically no technical novelty in this paper, since it's an application of an existing system (Larsen 2016) to volumetric data by changing the convolutional layers to volumetric convolutional ones. Although the paper is rather clear (in part due to its large technical overlap with existing work), there is a couple of details that could improve the clarity of the paper, namely: - In the object classification experiment in 4.2, the authors use the learned representation as features for classification, but what classifier is used? Is it the same for all algorithms? - Since the authors emphasize the probabilistic nature of the latent space, I think it would be useful to report how likely the interpolations in figure 6 are. Specifically, I'm wondering about how likely interpolations with broken components as arms or legs in chairs are. - The idea of analysing the effect of changing some dimensions in the latent representation, or analysing the neuron activation depending on the volume input is good, but the papers show too few results. A more through analysis shown in a video would be interesting. Overall, I think this paper brings good existing ideas from images to volumetric data, which pushes the state of the art in that field. However, the limited technical novelty reduce the potential impact of the paper.

Confidence in this Review

2-Confident (read it all; understood it all reasonably well)


Reviewer 4

Summary

This paper proposes to combine Generative Adversarial Network with Volumetric Convolutional Network to perform 3D shape generation. The combined network, called VAN, learns a latent representation for the objects, which is also used to generate shape. In the experiment section, the shape generation pipeline performs reasonably well and produces very convincing results. The experiment also shows that the latent representation z is semantically meaningful.

Qualitative Assessment

I think this paper presents an straightforward way to generate 3D objects. The idea of training a Generative Adversarial Network combined with a Volumetric Network to generate 3D shape is definitely novel. Furthermore, incriminating an image encoder to VAN in order to output the latent representation vector z from an image is interesting. The experiments are extensive and well-designed, and prove the effectiveness of the method. However, it gives me the impression that the method proposed is a combination of existing methods but applied to a novel problem setup, so the novelty is limited. Here are my concerns: 1. The paper argues that the shape generated by VAN is similar but not identical to the nearest neighbor retrieved from the training set. It’s very hard to judge the similarity by just looking at Figure 2., and thus not very convincing. They looks kind of the same to me. Can you provide any quantitative evaluation? 2.It’s not clear what exactly you did with the output feature from the second to last layer in Section 4.3.Do you use these features to retrain a classifier to perform classification? 3.What is the intuition behind the loss function of VAE-VAN in Equation (2) ? How do you train the VAE-VAN? Is the training procedure the same as training the VAN? 4.In section 4.3, the paper shows an experiment on reconstructing 3D from single images using VAE-VAN. Figure 4 shows a large variety of examples of furnitures taken at widely different angles. Does that mean the method is invariant to viewing angle? Does the images of the same object taken at different angles produce the same latent representation z, as well as the final shape? 5.The paper performs classification using the output of the second to last layer of the discriminator network as feature, and the results looks very promising. But given that the latent representation z is so meaningful as shown in section 5, have you tried to use z as a feature to do classification? I wonder what the result would be. 6.The output mesh grid looks very coarse, and I guess that’s because the low resolution you are using. In that sense, the network is actually learning some general mid-level structures of the shapes, like armpits or legs of a chair, just as it is demonstrated in your experiment. Have you tried a higher resolution? Does this method work on higher resolution, where there are more details? Can it still learn as well? Overall, I think this is an interesting paper. Please address my concerns.

Confidence in this Review

2-Confident (read it all; understood it all reasonably well)


Reviewer 5

Summary

This paper tackles the problem of 3D object generation. The method involves building a generative deep network, that is trained using the adversarial signal obtained from a counterpart discriminative deep network. GANs have shown to perform very well on generating 2D images, but this paper goes on to show that adversarial training can help in 3D object synthesis as well. The second extension that the paper proposes is to extend VAE-GANs to 3D, which the authors call VAE-VAN. The VAE-VAN model has an additional encoder network that encodes a 2D image into a latent representation, and the generative network which then reconstructs a 3D object model of the object present in the 2D image from the encoded latent representation. Experiments show that the proposed models are able to synthesize objects in 3D that look better than the state of the art, qualitatively. There are also quantitative experiments that show that the methods perform better than existing approaches.

Qualitative Assessment

This reviewer feels that the methods in the paper extend well-known techniques from 2D image generation to 3D object generation. Adversarial training and VAE-GANs have been proposed by others in the community. This work uses these two existing techniques, extend them to 3D synthesis, and show that they perform well. So, this work's contribution lies in the extension of the existing techniques to 3D, which is not trivial to do, and at the same time, not significantly novel. However, I do feel that the work is in the right direction and is of interest to the NIPS community. The reviewer does have some questions that should be made clear by the authors. I am a bit confused by the experimental details with regards to 3d object classification and single image 3d image reconstruction. Line 172 says that different VANs were trained for different object categories. If so, which discriminator is used to extract the representations (L193) for classification. Are separate VANs trained for the experiments in Sec 4.2 or is it a single VAN? In either case, what kind of classifiers are used to make the decisions (nearest neighbor, multi-class, etc.)? I have similar questions with the experiments in Sec 4.3? How many VAE-VANs are trained on the IKEA dataset? If there are multiple models trained (one for each object category), how are the models tested on novel test images? Also more analysis is required on understanding what each neuron in the discriminator responds to. From Fig 8. there seem to be multiple neurons that get activated by the same object part. The neurons corresponding to top-right and bottom-left both like left sides of chairs. I would like to see more experimental details to clarify these issues and to enable easy reproducibility.

Confidence in this Review

3-Expert (read the paper in detail, know the area, quite certain of my opinion)